# Effects of activity-oriented physiotherapy with and without eye movement training on dynamic balance, functional mobility, and eye movements in patients with Parkinson's disease: An assessor-blinded randomised controlled pilot trial

**Sarah Mildner[1], Isabella Hotz[1], Franziska Kübler[1], Linda Rausch[2], Michaela Stampfer-Kountchev[3], Johanna Panzl[3], Christian Brenneis[3,4], Barbara Seebacher** [1,4,5] *

**1** Department of Rehabilitation Science, Clinic for Rehabilitation Münster, Münster, Austria, **2** Department of Sport Science, University of Innsbruck, Innsbruck, Austria, **3** Department of Neurology, Clinic for Rehabilitation Münster, Münster, Austria, **4** Karl Landsteiner Institute for Interdisciplinary Rehabilitation Research, Münster, Austria, **5** Clinical Department of Neurology, Medical University of Innsbruck, Innsbruck, Austria

* barbara.seebacher@reha-muenster.at

## Abstract

### Objectives

To describe changes in balance, walking speed, functional mobility, and eye movements following an activity-oriented physiotherapy (AOPT) or its combination with eye movement training (AOPT-E) in patients with Parkinson's disease (PD). To explore the feasibility of a full-scale randomised controlled trial (RCT).

### Methods

Using an assessor-blinded pilot RCT, 25 patients with PD were allocated to either AOPT or AOPT-E. Supervised interventions were performed 30 minutes, 4x/weekly, for 4 weeks, alongside inpatient rehabilitation. Outcomes were assessed at baseline and post-intervention, including dynamic balance, walking speed, functional and dual-task mobility, ability to safely balance, health-related quality of life (HRQoL), depression, and eye movements (number/duration of fixations) using a mobile eye tracker. Freezing of gait (FOG), and falls-related self-efficacy were assessed at baseline, post-intervention, and 4-week follow-up. Effect sizes of 0.10 were considered weak, 0.30 moderate, and ≥0.50 strong. Feasibility was assessed using predefined criteria: recruitment, retention and adherence rates, adverse events, falls, and post-intervention acceptability using qualitative interviews.

**Data Availability Statement:** The datasets used and/or analysed during the current study are included in this published article.

**Funding:** The author(s) received no specific funding for this work.

**Competing interests:** The authors have declared that no competing interests exist.

## Results

Improvements were observed in dynamic balance (effect size r = 0.216–0.427), walking speed (r = 0.165), functional and dual-task mobility (r = 0.306–0.413), ability to safely balance (r = 0.247), HRQoL (r = 0.024–0.650), and depression (r = 0.403). Falls-related self-efficacy (r = 0.621) and FOG (r = 0.248) showed varied improvements, partly sustained at follow-up. Eye movement improvements were observed after AOPT-E only. Feasibility analysis revealed that recruitment was below target, with less than two patients recruited per month due to COVID-19 restrictions. Feasibility targets were met, with a retention rate of 96% (95% confidence interval [CI]: 77.68–99.79) and a 98.18% (95% CI: 96.12–99.20) adherence rate, exceeding the targets of 80% and 75%, respectively. One adverse event unrelated to the study intervention confirmed intervention safety, and interview data indicated high intervention acceptability.

## Conclusions

AOPT-E and AOPT appeared to be effective in patients with PD. Feasibility of a larger RCT was confirmed and is needed to validate results.

## Background

Parkinson's disease (PD) is the second most common neurodegenerative disease with a prevalence of 1–2 per 1000 of the population, and 1 percent of people older than 60 years [1]. Cardinal symptoms of PD are akinesia, rigidity, tremor at rest and postural instability in varying degrees [2]. Studies show that people with PD are generally slower to respond [3, 4] as a result of increased inhibition of the basal ganglia [5]. In addition, oculomotor dysfunction has been described including reduced saccade velocity and higher saccadic latencies, which may contribute to limitations in the perception of the environment [6–8]. Other studies have shown longer visual response times, with a higher number of fixations and square wave jerks, suggesting fixation stabilisation difficulties [9, 10]. Gaze-holding abnormalities in patients with PD were thus characterised by unusually excessive fixational eye movements similar to microsaccades [11]. In combination with bradykinesia, changes in saccade parameters, particularly saccadic hypometria [10], may increase the risk of falls in daily life [12].

Due to a lower saccade amplitude and slower eye movement speed [5, 7], there is also an eye movement limitation particularly during turning [13, 14]. Therefore, people with PD are more likely to fall than adults of the same age, especially while turning [15]. In the treatment of people with PD, visual dysfunctions should be addressed appropriately to promote coordination and reduce the risk of falling [13]. Falls are associated with limitations in independence in activities of daily living, which negatively affect the participation in society of people with PD [16].

Research has shown that people with PD have lower saccade frequency compared to controls [17, 18]. During turns, saccade frequency increases in both groups, while it decreases under dual-task conditions [17]. In PD, attention deficits directly relate to reduced saccade frequency, impaired visual function, and gait impairment, with significant implications for falls [17]. Although saccade frequency does not directly impact gait impairment in PD, it is indirectly influenced by visual function deficits, which affect gait through fluctuating attention [17, 18]. This highlights the importance of visual observation in maintaining safe gait and suggests a need for increased awareness of environmental scanning frequency for overall safety [17].

Although previous studies have shown that eye movement training combined with postural control and/or gait training may be successful in patients with progressive supranuclear palsy (PSP) [19], no studies on patients with PD have been conducted so far. A published protocol of 2019 investigated the improvement in voluntary saccade function using voluntary saccade training, but the results have not been reported yet [20]. Furthermore, in order to ascertain the potential impact of specific eye movement training on improving the efficacy of eye movements in patients with PD, we intended to explore the number of whole fixations on predetermined targets and their overall visual field.

The aims of this study were to investigate the preliminary effects of activity-oriented physiotherapy combined with and without an eye movement training on dynamic balance, functional mobility, and eye movements in patients with PD. Further aims were to evaluate the feasibility of a full-scale randomised controlled trial (RCT) using pre-defined feasibility criteria of recruitment, retention and adherence rates, adverse events, and the acceptability of the intervention.

## Materials and methods

### Study design and setting

The study was performed at the Department for Neurology, Clinic for Rehabilitation Münster, Austria. A single-blinded parallel design pilot RCT was conducted to investigate the feasibility and preliminary effects of an activity-oriented physiotherapy with and without eye movement training on dynamic balance, functional mobility, and eye movements in patients with PD (see S1 File for a CONSORT Checklist). With semi-structured interviews on the acceptability of the intervention in the experimental group only, a mixed methods approach was pursued. Qualitative methods are described in S2 File. Ethical approval was gained from the research ethics committee of the Medical University of Innsbruck (reference 1038/2021; date: 01.04.2021). Written informed consent was obtained before enrolment from all participants who could withdraw from the study at any time without treatment prejudice. The collected data were pseudonymised and treated confidentially. This study was prospectively registered with the German Clinical Trials Register on 08.04.2021 (DRKS-ID: DRKS00024982).

### Participants and randomisation

**Inclusion criteria.** Patients with PD were screened for eligibility by neurologists. Men and women with PD according to the UK Brain Bank criteria [21], aged 30–80 years, of any ethnicity, able to walk without assistance, German speaking and writing, with adequate cognitive function (Mini Mental State Examination [MMSE] [22] score ≥24/30), Hoehn & Yahr (H&Y) scale score 1–3 [23] during the ON medication phase and stable dosage of dopaminergic therapy at least 3 weeks before the start of the study or without dopaminergic treatment for the duration of the study were included.

**Exclusion criteria.** Patients with concomitant diseases (e.g., malignant diseases, other neurological, orthopaedic, cardiac, or psychiatric disorders, major depression, dementia), photosensitivity, non-parkinsonian gait disorder (e.g., due to musculoskeletal symptoms), recent surgery (general and eye surgery), intraocular implants, strabismus, nystagmus, severe drooping eyelids, untreated pain, uncorrected visual or hearing impairment, which may be a barrier to the training or assessments, pregnancy, recent deep brain stimulation (DBS) or a change in DBS parameters within the previous year, severe motor fluctuations were excluded. Initiation of a new dopaminergic medication or adjustment thereof within the study period was not planned. As this is a study in a real-life setting, the principal investigator was allowed to decide whether an adjustment was necessary, however.

**Sample size.** Given the pilot nature of this study, a formal sample size calculation was not performed. Instead, we relied on guidance from previous research. Julious recommended a minimum of 12 participants per group for pilot studies [24] whereas Browne suggested 30 participants for a two-arm study [25]. Considering potential attrition, our goal was to recruit 17 patients per group.

**Recruitment and randomisation.** Eligible patients were informed about the study orally and in writing by the principal investigator and were invited to participate in the study. After having obtained written informed consent, screening for cognitive impairment was performed. If patients passed the screening, they were randomly assigned into one of two groups (activity-oriented physiotherapy with eye movement training [AOPT-E] or activity-oriented physiotherapy [AOPT]) using a 1:1 ratio and random number sequence generated using an online software (Sealed Envelope Ltd, London, United Kingdom). Stratified block randomisation, with blocks of 4 and 6 was conducted by an independent researcher not involved in participant recruitment, intervention instructions or testing. Due to the small sample size, stratification was based only on levels of impairment (H&Y levels 1; 2–3), a relevant predictive factor for a change in dynamic mobility [26]. Allocation was concealed from the principal investigator and neurologists, with sequentially numbered, sealed opaque envelopes being used.

## Intervention

Participants, admitted to the clinic primarily for PD, received the study intervention alongside inpatient rehabilitation at a specialist PD rehabilitation ward. This comprehensive programme included various therapeutic modalities such as medical training therapy, balance and gait training, strength and endurance exercises, occupational therapy, speech and language therapy, dietary guidance, psychological counselling, and patient education. Both intervention groups received 16 intervention sessions of 30 minutes each for 4 weeks. Study participants were supervised 1:1 by a trained and experienced neurological physiotherapist. AOPT-E and AOPT were delivered in a darkened room with white light at 300 lux. In both the AOPT-E and AOPT groups, the activity-oriented physiotherapy intervention was based on motor learning principles [27], evidence-based and aligned with guidelines for PD rehabilitation [16, 28, 29]. Exercises in the AOPT-E and AOPT groups aimed to improve balance, functional mobility, walking, strength, and coordination. Each session consisted of warm-up exercises, the main part, and cool-down exercises. In both groups, the exercises were performed in different starting positions (seated, wide stance, tandem stance, etc.) to practice a variety of movements and situations close to everyday life such as alternating large and small steps, walking backwards, or lunges. Physiotherapy interventions involved balance, dual-task, coordination, functional mobility, walking, and stair climbing exercises and were identical in both groups except for the additional eye movement training in the AOPT-E group. Exercises involved hand-eye coordination activities like tossing and catching a ball, coordination hoop, rice bags, or bean bags. Exercises were gradually increased in difficulty according to the patient's performance level. A total of 56 and 45 exercises were performed in the AOPT-E and AOPT group respectively (see S1 Table for a template for intervention description and replication (TIDieR) Checklist [30]).

In the AOPT-E group, activity-oriented physiotherapy was combined with eye movement training. The focus of the eye movement training was on the saccade size and number of fixations and (increasing) the visual field. For the design of the eye movement training, the parameters of exercise intensity, duration, performance during the ON-phase, and distance of the participants from the exercise poster were derived from previous studies [20, 31] and adapted to the purpose of the present study. For eye movement training, a rectangular exercise poster

was placed on the wall at a height of 0.9 metre and a 0.7 metre distance from the participant. The participant was instructed to keep both feet on the floor, sit as upright as possible and move the head as little as possible during the poster eye movement training (the dimensions and a photo of the poster are shown in S3.1 Fig in S3 File). The training commenced with an eye warm-up to minimise the risk of injury. Eye movement training involved performing visual search tasks from the poster centre outwards either in a fixed or random order. We aimed to alternate saccades and fixations i.e., during the training patients were instructed to direct their gaze towards specific colour dots embedded in the poster, which also included several differently coloured distractor dots. Each eye movement began with a black coloured central fixation dot. Of the 16 dots, 4 were in the corners of the poster and 3 each were aligned on a vertical or horizontal axis from the central fixation point to the top, bottom, left, and right. Additionally, walking training was performed on a hallway with wall markings; using a crosshair through adhesive tape and adhesive dots on the wall (S3.2 Fig in S3 File) participants were asked to perform various balance, coordination, and functional mobility exercises, combined with visual fixation or search tasks utilising the wall markings.

## Data collection

Demographic and disease-related data were collected from patients' medical charts at baseline. Outcome data were collected at baseline and 4-week post-intervention at the study centre and at 4-week follow-up via telephone interview. Physicians and physiotherapists were blinded to the group allocation of patients. Therapists were not blinded as they had to be aware of the exercise programme of the respective group.

**Co-primary outcomes.** Co-primary outcomes were dynamic balance during walking as assessed by the 10-items Functional Gait Assessment (FGA) [32] and eye movements during the FGA items 5, 6, and 10 using a mobile eye tracker (Tobii Pro Glasses 3). For FGA scoring, a 4-point ordinal scale ranging from 0 = severe impairment to 3 = normal ambulation was utilised [32]. For people with PD, the cut-off value of the FGA is ≤18/30 points for identifying an increased risk of falls [33].

**Secondary outcomes.** Secondary outcomes were walking speed as assessed by the 10-Metre Walk Test (10MWT) [34], functional and dual-task mobility using the Timed Up and Go (TUG) [35] and TUG with a motor task (TUGman) [36] respectively. Further secondary outcomes were the ability to safely balance as measured using the Berg Balance Scale (BBS) [37], and dynamic balance and coordination using the Four Square Step Test (FSST). The number of falls [38] over 3 months prior to the study, during the 4-week intervention and 4-week follow-up periods was recorded. Other secondary outcomes were health-related quality of life as assessed with the Parkinson's Disease-39 Questionnaire (PDQ-39) [39], freezing of gait (FOG) using the Freezing of Gait Questionnaire (FOGQ) [40], falls-related self-efficacy with the Falls Efficacy Scale-International (FES-I) [41] and depression with the revised Beck Depression Inventory (BDI-II) [42]. For all questionnaires, the validated German versions were used. The follow-up call was used to collect data on the fall rate during the 4-week follow-up period, FOG (FOGQ) [40] and falls-related self-efficacy (FES-I) [41]. During eye movement data collection, patients wore eye-tracking glasses and an audio-recorded standardised instruction for a visual search task was used for guidance.

**Feasibility criteria.** The feasibility criteria for conducting a larger RCT were defined a priori [43] and included the following:

a. a target recruitment rate of 40% of 85 eligible patients (or 4–5 participants per month); the total number of eligible PD patients was estimated based on the number patients with PD treated at the study centre in the last 3 years.

b.  a target retention rate of 80%,

c.  a minimum target adherence rate of 75% (at least 3 of 4 scheduled intervention sessions per week),

d.  high safety of the intervention, no severe adverse events, or only very mild and transient adverse events (monitored throughout the study),

e.  high acceptability of the intervention, evaluated through semi-structured individual interviews in the AOPT-E group (S2 File).

**Eye-tracking.**   As a further secondary outcome, using an objective instrument for measuring eye movements, mobile eye-tracking was used at baseline and post-intervention. Tobii Pro Glasses 3 (Tobii AB, Stockholm. Sweden) were utilised to register eye movements and fixations, with 16 illuminators and 4 eye camaras integrated, a sampling rate of 50 Hertz and a scene camera with a wide field of view (diagonal: 106-degree, horizontal: 95-degree, vertical: 63-degree).

Initially, patients were informed about the use of eye-tracking glasses and possible side effects such as temporary dizziness. Eye movement assessment was done utilising the rectangular training posters and followed the same principles as the training. The fixed-order and random-order fixation sequences during the measurements were standardised. The participant was instructed to keep both feet on the floor, sit as upright as possible and move the head as little as possible during the poster eye movement assessment (S3.1 Fig in S3 File). In addition, eye movements were tracked during the FGA tasks No. 5 "gait and pivot turn" and No. 6. "stepping over an obstacle".

## Statistical data analysis

Statistical data analysis was performed using IBM SPSS software, version 28.0 (IBM Corporation, Armonk, NY, USA) and GraphPad Prism 9, San Diego, California. An attempt was made to avoid missing data by inspecting the questionnaires after completion and, in the case of unanswered questions, asking participants to complete them.

Descriptive statistics were used for the demographic and disease related data, primary and secondary outcome variables. Counted and nominal data (sex; FOG; fall rate, recruitment, retention, and adherence rates) were reported as absolute and relative frequencies (N, %) and ordinal variables (MMSE, UPDRS, H&Y, FGA, BBS, PDQ-39, FOGQ, FES-I, BDI-II) using the median (25th and 75th percentiles). Normality of data distribution of continuous variables (age, disease duration, TUG, TUGman, FSST, 10MWT) was checked using the Shapiro-Wilk test and visual inspection of histograms and data presented using mean (standard deviation [SD]) or median (25th; 75th percentiles) as appropriate. In this small-scale pilot study, we calculated the change scores from baseline to post-intervention for primary and secondary outcomes. Hence, we did not conduct hypothesis testing to assess treatment effects within or between groups [44]. The effect size calculation was derived from a Mann Whitney-U test, which was conducted on transformed variables representing differences between baseline and post-intervention measures. The standardized U-value and the total number of observations (n) underlying Z were utilised in the calculation: r = Z/sqrt(n) [45]. According to Cohen (1988), a correlation coefficient of 0.10 is considered weak, 0.30 is considered moderate, and 0.50 or higher is considered strong. [46].

**Feasibility analysis.**   Estimation of the eligibility, recruitment, and adherence rates was done according to the following equations:

$$\text{Eligibility rate} = (N_{\text{Eligible}}/N_{\text{Examined}})*100 \tag{1}$$

$$\text{Recruitment rate} = (N_{\text{Consent}}/N_{\text{Eligible}})*100 \tag{2}$$

$$\text{Retention rate} = (N_{\text{Completed}}/N_{\text{Consented}})*100 \tag{3}$$

$$\text{Adherence rate} = (N_{\text{Performed}}/N_{\text{Total}})*100 \tag{4}$$

where $N_{\text{Examined}}$ is the number of people who were examined for eligibility for this study; $N_{\text{Consented}}$ is the number of patients who signed the informed consent; $N_{\text{Eligible}}$ is the number of patients who met the inclusion and exclusion criteria, $N_{\text{Completed}}$ is the number of people who completed the study; $N_{\text{Performed}}$ is the number of performed therapy sessions; and $N_{\text{Total}}$ is the number of planned therapy sessions.

The eligibility, recruitment and adherence rates were calculated using the Wilson 'score' method propagated by Newcombe and its 95% confidence interval (95% CI) [47]; in the case of a proportion close to 0 or 1, a Poisson approximation according to Brown was used [48].

**Eye movement analysis.** Eye movements were analysed using TobiiProLab (v.1.145) analysis software. Data on the fixations duration (milliseconds), and number of fixations were exported. First, the relevant intervals (areas of interest) were selected, then fixations detected by the software were automatically created as an image. These fixations on the image were then manually checked by the researcher for accuracy and presence to improve the quality. To make the fixations more understandable in terms of frequency and location on the poster, snapshots of the fixations were taken. The dots were then marked as Areas of Interest (AOI), which could be evaluated separately. For the FGA tasks, the field of view was grouped into AOIs to allow tracking of eye movements and fixations in different areas (e.g., turning task: left, centre, right, bottom/floor, top/straight). The video was then exported using the AOI-based filter. The results of the exported metrics were manually checked and verified. Eye movement data was plotted using bubble plots. In addition, gaze plots were used to visualise the eye tracking data using the wall poster. Gaze plots show the location, number, sequence, and duration of gazes at specific locations. Initially, an "assistant mapping" was conducted, followed by manual checks for precision and consistency by the researchers.

## Results

The recruitment of patients occurred between 25.04.2021 and 26.09.2022. Due to COVID-19 restrictions, safety measures and staffing limitations, recruiting the intended 34 patients with PD was not feasible. Out of the 25 randomised participants, 24 successfully completed the study and were included in the analysis (AOPT-E 12, AOPT 12), corresponding with a 4% attrition rate. A CONSORT flow diagram (extension to randomised pilot and feasibility trials [49]) is presented in Fig 1.

### Participants' characteristics

Ten women and fourteen men with a mean (standard deviation [SD]) age of 67.4 (11.3) years and mean (SD) disease duration of 5.5 (4.1) years completed the study (Table 1). Their median (minimum—maximum) UPDRS score was 32 (15–59) and the median Hoehn & Yahr scale was 2.0 (1.0–3.0). At baseline, groups were comparable with respect to demographic and disease related characteristics, primary and secondary outcomes.

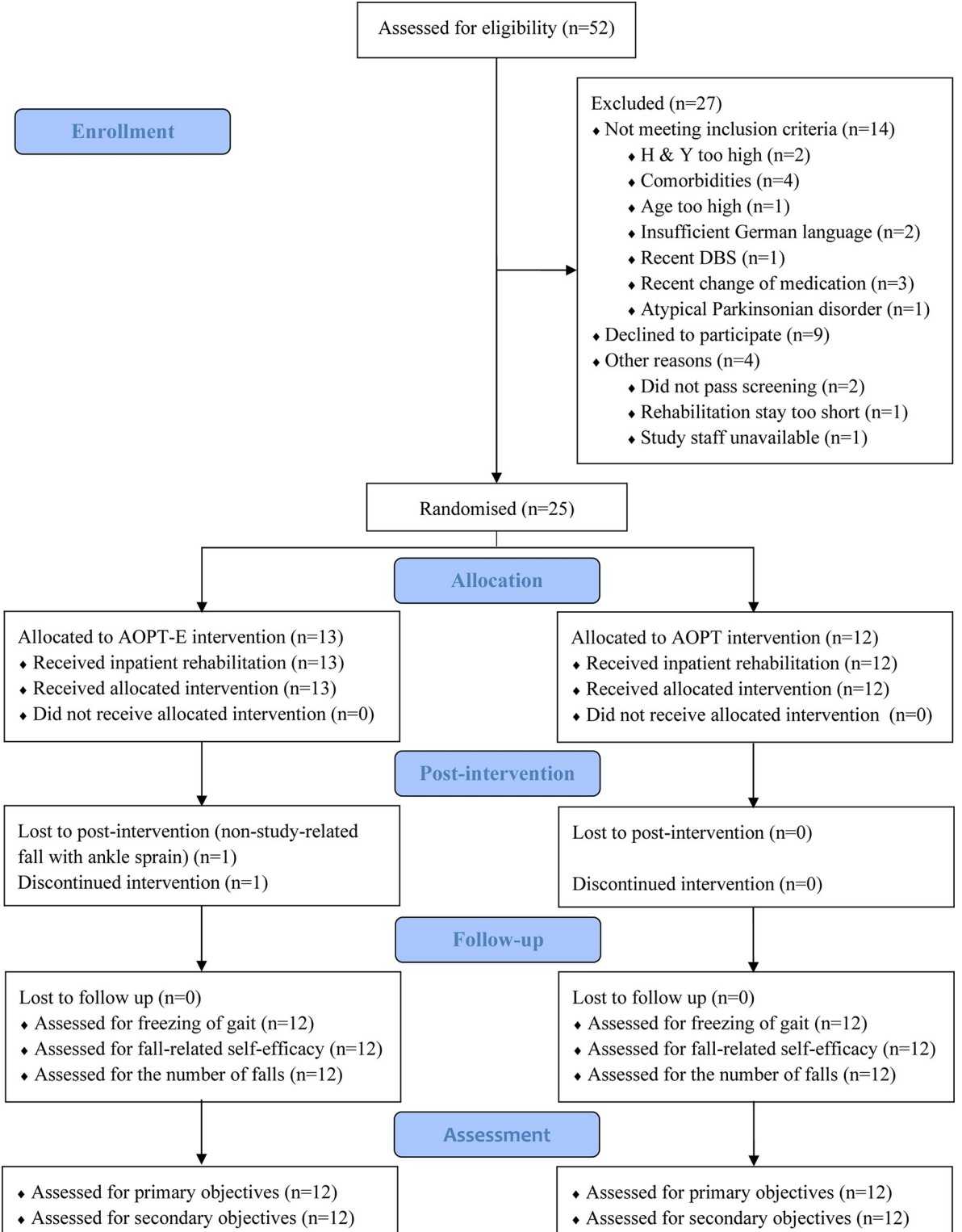

**Fig 1. CONSORT flow diagram.** DBS: deep brain stimulation; H&Y: Hoehn & Yahr scale.

Table 1. Participants' demographic and disease-related characteristics.

| | AOPT-E group | AOPT group |
|---|---|---|
| | N = 12 | N = 12 |
| Sex[1] (female: male) | 7: 5 | 3: 9 |
| Age[2] | 67.9 (8.5) | 66.9 (13.9) |
| Disease duration[2] | 6.0 (4.0) | 5.0 (4.4) |
| Hoehn & Yahr scale[3]* | 2.0 (1.0–3.0) | 2.0 (1.0–3.0) |
| Freezing of gait[1] (yes: no) | 7: 5 | 4: 8 |
| Dopa equivalence dosage (mg)[2] | 499.6 (322.5) | 563.6 (357.8) |
| Mini Mental State Examination[3]** | 29.0 (25.0–30.0) | 28.0 (24.0–30.0) |
| Unified Parkinson's Disease Rating Scale* | 34.0 (15.0–59.0) | 30.0 (18.0–59.0) |
| Beck Depression Inventory-II* | 11.0 (2.0–22.0) | 8.0 (11.0–13.0) |

[1]Frequencies

[2]Mean (standard deviation)

[3]Median (Minimum—maximum).

*Higher scores represent worse functioning

**Higher scores represent better functioning.

AOPT: activity-oriented physiotherapy; AOPT-E: activity-oriented physiotherapy with eye movement training; N: number of participants.

## Co-primary outcomes

At baseline, a median (minimum–maximum) FGA score of 22.0 (4.0–28.0) out of 30 was observed in the total study sample, with 8 patients scoring between 4.0 and 18.0. According to the cut-off value ≤18 points, these 8 patients were identified as being at increased risk of falls. There was an improvement in dynamic balance during walking from baseline to post-intervention as assessed using the FGA for both the AOPT-E and AOPT groups (Fig 2 and S2 Table).

For the analysis of eye movements during FGA item No. 5 "pivot turn during walking", the field of vision was divided into three horizontal thirds, where 1 stands for the first third in the direction of rotation and 2 for the middle third and 3 for the last third. In addition, it was measured whether the eye fixation was in the upper (straight) or lower (floor) half. Both the duration and number of fixations are shown darker the more frequent or longer the fixation was in that area. The analysis revealed that participants in both groups were not able to fully direct their gaze towards the turning direction as their visual focus did not extend to the farthest right visual region. There were no relevant changes seen from baseline to post-intervention in any group. With respect to directing gaze towards the ground, after the intervention, participants in the AOPT-E group exhibited a decrease in the maximum duration of fixation, whereas those in the AOPT group displayed an augmentation in the maximum duration of fixation (Fig 3).

The analysis of FGA item No. 6, involving "stepping over an obstacle," indicated that the AOPT group maintained a relatively consistent level of fixations post-intervention. In contrast, the AOPT-E group exhibited a reduction in both the duration and quantity of fixations on the ground prior to the stepping-over action, while displaying an increased number of fixations on the object and the ground behind it before the action. While performing the obstacle-crossing task, participants directed their gaze toward the ground less frequently, implying an expanded visual field and enhanced dynamic balance while walking (Fig 4).

Regarding FGA item 10, we faced technical challenges that hindered our ability to gather accurate data. Inadequate lighting conditions along the stairway and ongoing difficulties with calibration during participants' stair climbing resulted in inaccuracies in the recorded gaze data. As a result, conducting a meaningful data analysis was not possible.

## Functional Gait Assessment

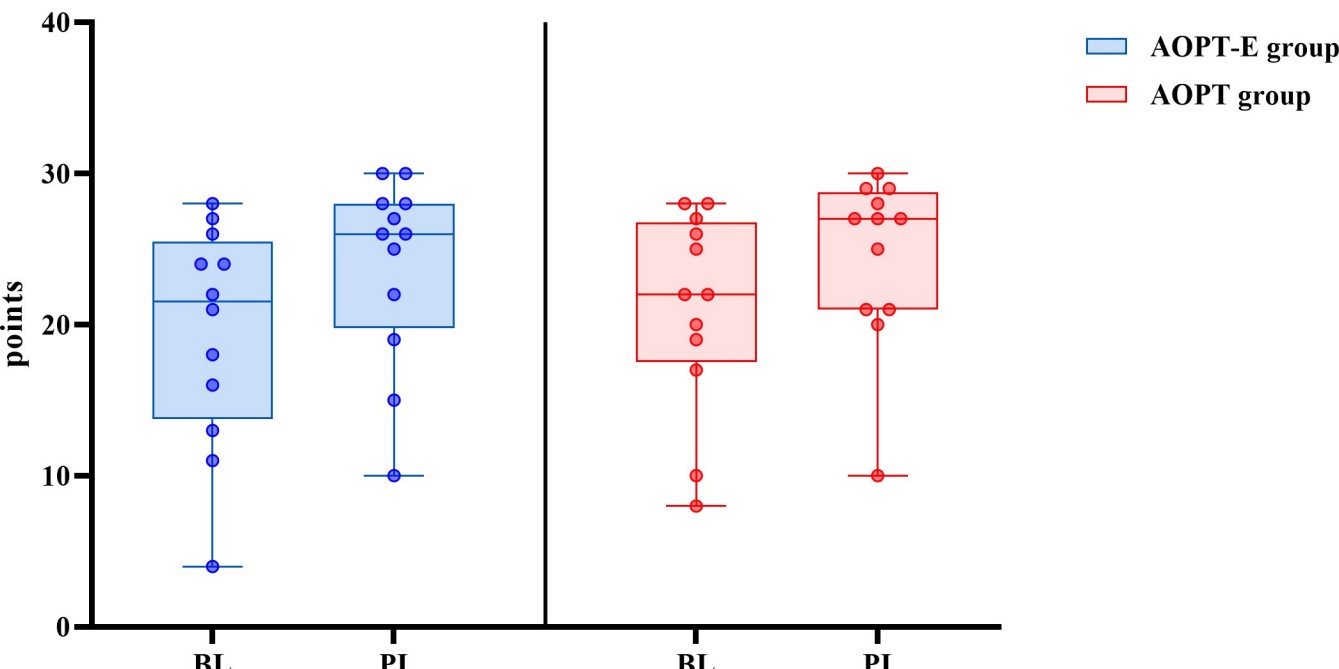

**Fig 2. Effect of interventions on dynamic balance during walking.** Box plots represent changes in dynamic balance during walking in the AOPT-E (left) and AOPT (right) groups, assessed by the Functional Gait Assessment (FGA). Medians are shown by lines in the centre of the boxes, the interquartile ranges are indicated by the boxes and ranges by the whiskers. Each dot represents one participant. BL: baseline, PI: post-intervention.

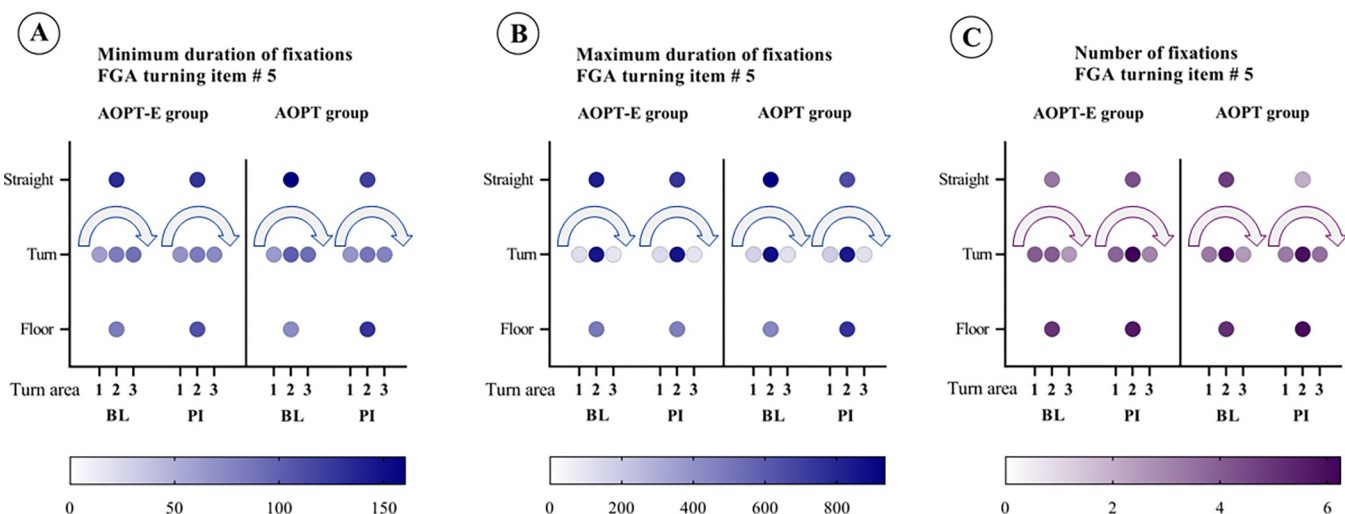

**Fig 3. Comparison of eye movements during pivot turning while walking between the two groups.** Bubble plots of eye movements during turning of the AOPT-E and AOPT groups: minimum duration of fixations in milliseconds at baseline and at post-intervention (A); maximum duration of fixations in milliseconds at baseline and post-intervention (B), and number of fixations at baseline and post-intervention (C). Darker colours represent a longer duration of fixations. AOPT-E: activity-oriented physiotherapy plus eye movement training; AOPT: activity-oriented physiotherapy; BL: baseline; FGA: Functional Gait Assessment; PI: post-intervention.

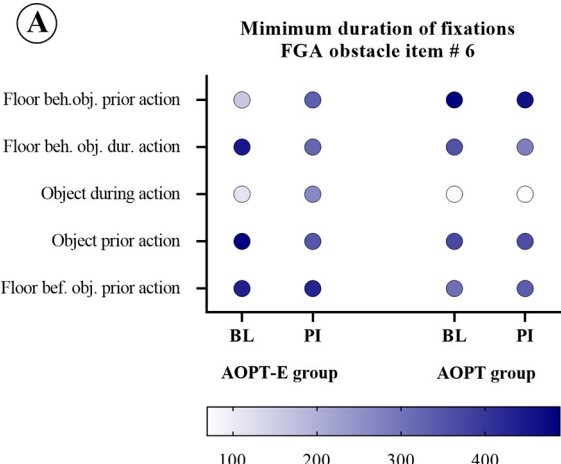

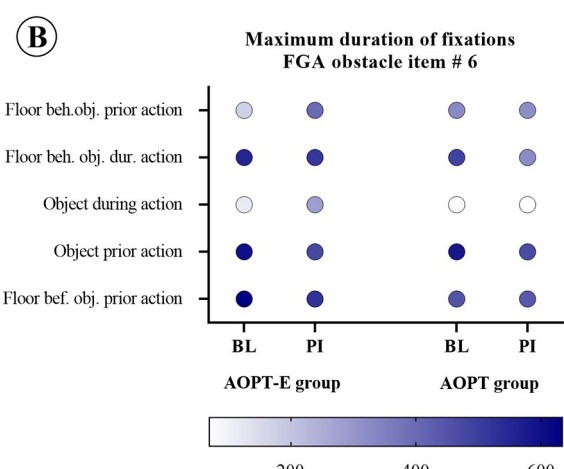

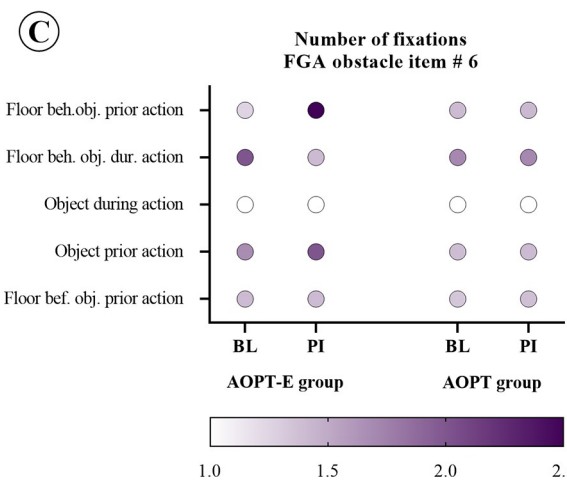

**Fig 4. Comparison of eye movements during stepping over an obstacle between the two groups.** Bubble plots of eye movements during stepping over an obstacle of the AOPT-E and AOPT groups: the minimum duration of fixations in milliseconds at baseline and at post-intervention (A); maximum duration of fixations in milliseconds at baseline and at post-intervention (B); and number of fixations at baseline and at post-intervention (C). Darker colours represent a longer duration of fixations and greater number of fixations respectively. Stepping over the obstacle is visualised by the

action. AOPT-E: activity-oriented physiotherapy plus eye movement training; AOPT: activity-oriented physiotherapy; bef.: before; beh.: behind; BL: baseline; FGA: Functional Gait Assessment; obj.: object; PI: post-intervention.

## Secondary outcomes

With respect to walking speed (10MWT), an improvement from baseline to post-intervention was observed for AOPT-E and AOPT, respectively (Fig 5 and S3 Table).

After AOPT-E and AOPT improvements in functional mobility (TUG), functional mobility with a motor task (TUGman), the ability to safely balance (BBS), and dynamic balance (FSST) were observed. Changes in functional mobility with a motor task and dynamic balance seemed greater in the AOPT-E group whereas changes in functional mobility and the ability to safely balance were similar in the two groups. Participants' fall rate over the past three months prior to start of the study was higher in the AOPT-E group, with 4 versus 0 falls in the AOPT group. After AOPT-E, the number of falls remained stable while following AOPT, there was an increase by 1 fall. At 4-week follow-up, participants in the AOPT-E group reported 2 falls, while numbers returned to zero in the AOPT group (Fig 6 and S3 Table).

Following 4-week AOPT-E improvements in the PDQ-39-assessed health-related quality dimensions of mobility, activities of daily living, emotional well-being, stigma, cognition, communication, and bodily discomfort were seen, but not social support. After AOPT, improvements in mobility and stigma, but no other HRQoL dimensions were observed (S4 Table).

With FOG (FOGQ), an improvement was observed after AOPT-E only, with reductions in FOG retained at 4-week follow up. In the AOPT group, at follow-up FOGQ scores exceeded

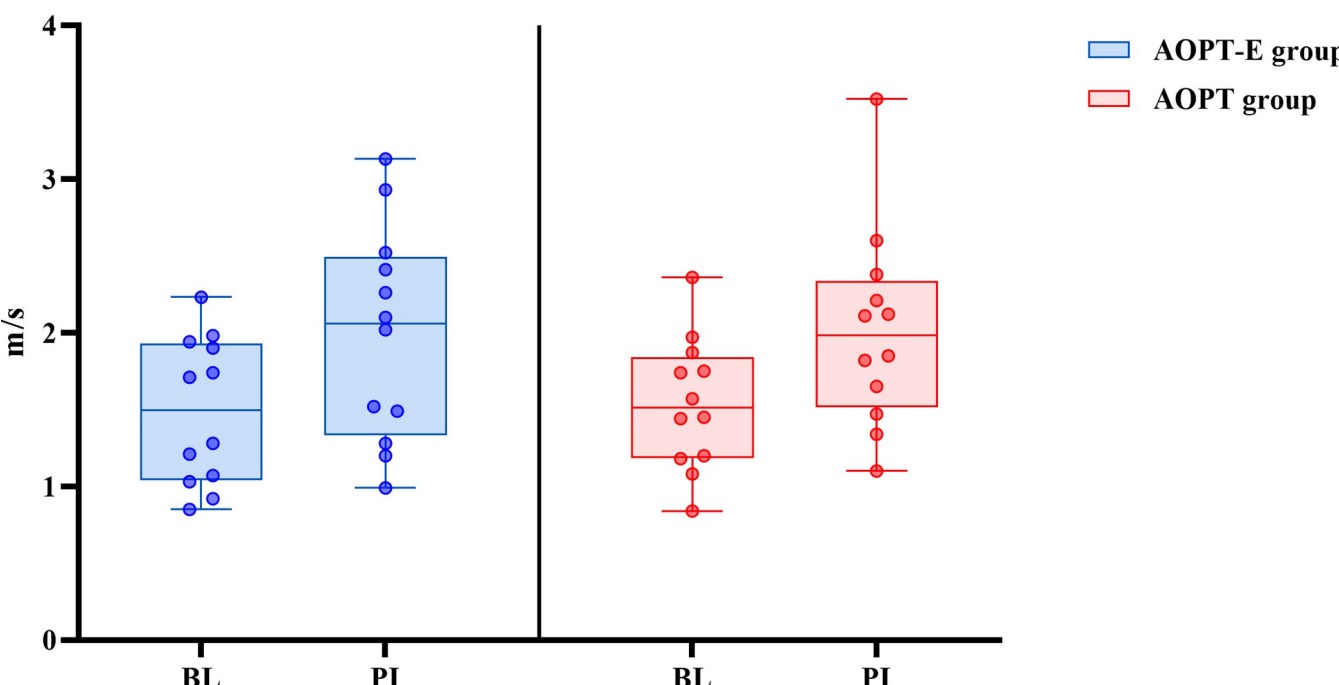

**Fig 5. Effect of interventions on walking speed.** Box plots represent changes in walking speed in the AOPT-E (left) and AOPT (right) groups, assessed by the 10-Metre Walk Test (10MWT). Medians are shown by lines in the centre of the boxes, the interquartile ranges are indicated by the boxes and ranges by the whiskers. Each dot represents one participant. BL: baseline, PI: post-intervention.

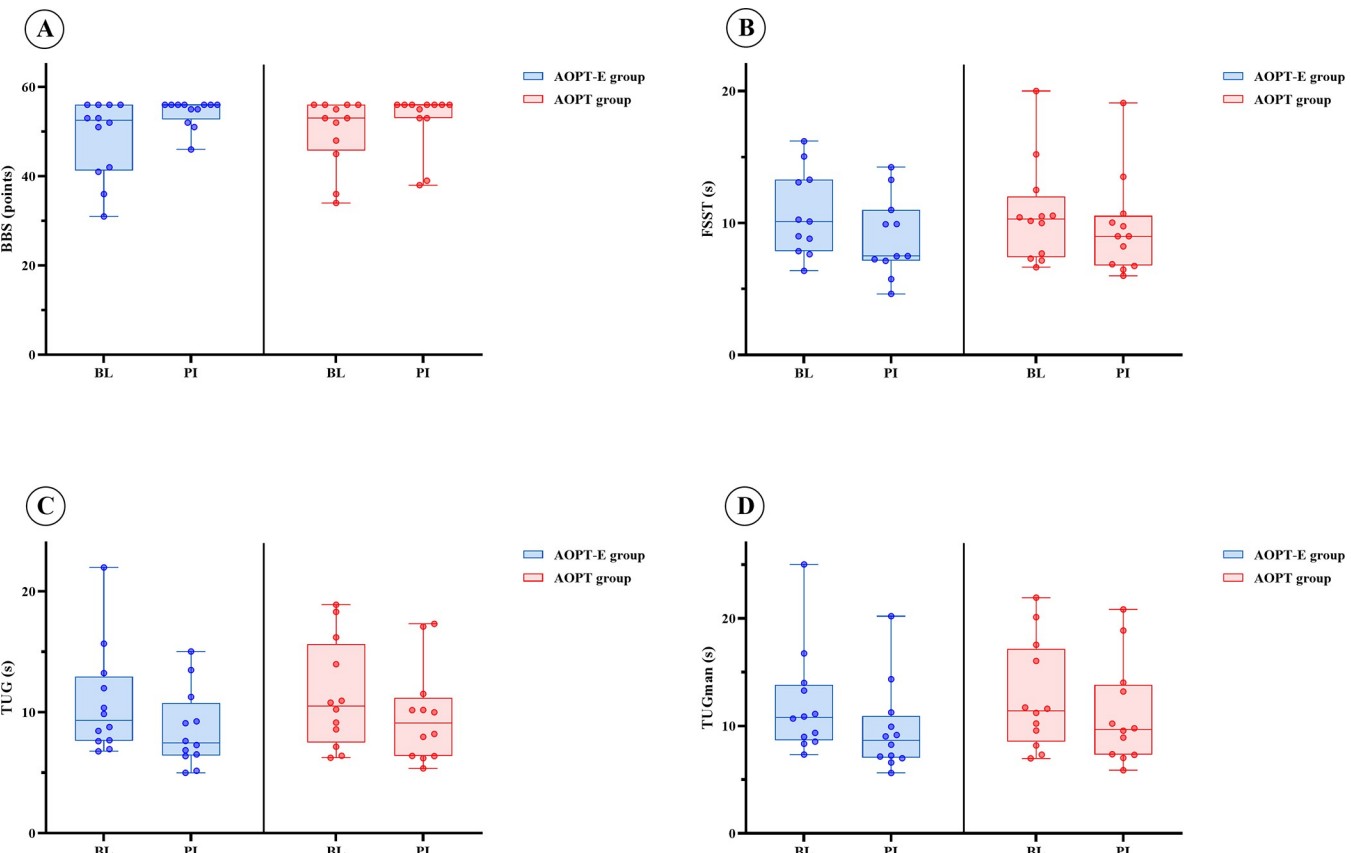

**Fig 6. Effect of interventions on balance and functional mobility.** Box plots represent changes in the ability to safely balance (Berg Balance Scale, BBS), dynamic balance (Four Square Step Test, FSST), functional mobility (Timed Up and Go, TUG), and functional mobility with a motor task (Timed Up and Go manual, TUGman) in the AOPT-E (left) and AOPT (right) groups. Medians are shown by lines in the centre of the boxes, the interquartile ranges are indicated by the boxes and ranges by the whiskers. Each dot represents one participant. BL: baseline, PI: post-intervention.

baseline values. After both interventions, there were improvements in falls-related self-efficacy (FES-I), which could be retained at follow-up in the AOPT-E group (S4 Table).

For depression (BDI-II), over the 4-week intervention period, improvements were found in both groups (S4 Table).

**Feasibility.** Feasibility analysis showed that of 39 eligible patients, 25 were enrolled in the study within 17 months, yielding a recruitment rate of 64.10 (95% CI 47.15–78.32). This means that less than two patients were enrolled in the study per month, and it was not possible to recruit 34 patients with PD, which does not meet the target. The recruitment delay was attributed to COVID-19 restrictions at the rehabilitation centre during the trial and potential overestimation of patient admission rates from 2018–2020. The eligibility rate was 73.08% (95% CI 58.74–84.00). The retention rate was 96% (95% CI 77.68–99.79) and the adherence rate was 98.18 (95% CI 96.12–99.20), indicating feasibility success. There was only one adverse event that was unrelated to the study intervention or procedures, confirming the safety of the intervention. With respect to the acceptability of the AOPT-E intervention, analyses of the interview data confirmed high acceptability. Five themes were identified through the thematic analysis. (1) Participants appreciate the study's organisation, varied content, and appropriate challenge level, expressing a need for breaks. (2) Participants find the training intense and suggest that future sessions could benefit from a more balanced approach, potentially incorporating rest days. (3) Participants prefer targeted patient education, a home exercise programme,

and progress feedback from therapists. (4) Main goals include increasing walking safety, coping strategies for the disease, active training opportunities, preferred training times, and a positive patient-therapist relationship. (5) Participants enjoy the intervention, reporting subjective improvements in daily activities, and find it meaningful (see S2 File for detailed findings). These themes underscore the feasibility of an eye movement training for people with PD.

## Eye tracking

At baseline, eye tracking analyses of standardised eye movement sequences using the poster indicated large inter-individual differences regarding participants' abilities to fix their gaze. Fig 7 illustrates two examples showing that the number of fixations and the accuracy related to the dots on the poster varied greatly.

Poster analysis of all AOPT-E group participants showed a decrease in the number of fixations from baseline to post-intervention and longer duration of fixations at the poster borders (edges, maximum right and left, top and bottom), indicating increased peripheral vision and overall visual field, and better eye movement control. This was even more pronounced with

**A**

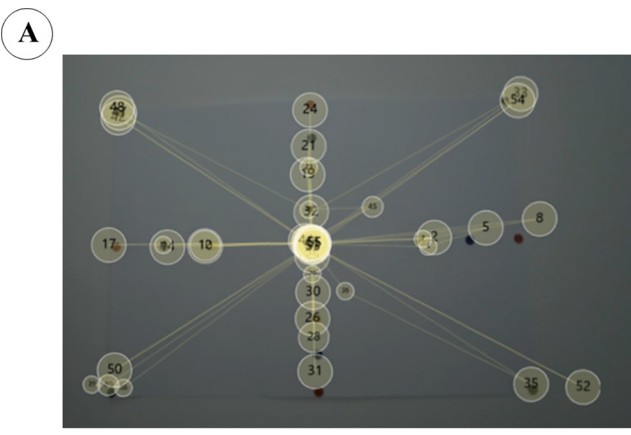

**B**

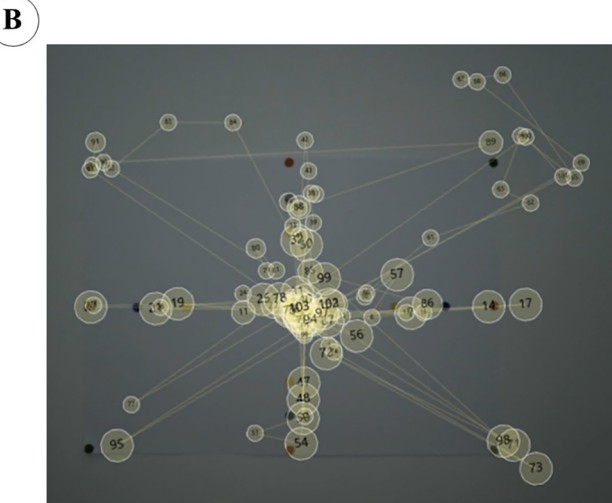

**Fig 7. Gaze plots of two selected participants.** Using the wall poster, gaze plots of participants illustrate normal eye movements in a patient with Hoehn & Yahr stage 1 (A) and dysfunctional eye movements in a patient with Hoehn & Yahr stage 3 (low accuracy, abnormally frequent and substantial square wave jerks alongside short duration of fixation on target dots). Larger circles indicate a longer duration of fixation on the respective dots.

the mixed poster, where the eye movement sequence was in random order (S3.3 Fig in S3 File).

## Discussion

This assessor-blinded pilot RCT aimed to investigate the preliminary effects of AOPT-E and AOPT on dynamic balance, functional mobility, and eye movements in patients with PD. Further aims were to evaluate the feasibility of a full-scale RCT using pre-defined feasibility criteria of recruitment, retention and adherence rates, adverse events, and the acceptability of the intervention.

We observed improvements after both AOPT and AOPT-E, with respect to the primary outcomes. These results were used to inform the sample size calculation of a full-scale RCT: Based on the primary outcome results (FGA), the sample size for a full-scale RCT was determined. Utilising G*Power version 3.1.9.7 [50], with an assumed power of 80%, a type I error probability of 0.05, a two-tailed test, and an effect size of r = 0.297, a total sample size of 282 participants is required to detect a true between-group difference. Accounting for a 15% attrition rate, a total sample size of 332 participants is recommended.

We further noted positive changes following both AOPT and AOPT-E interventions in most of the secondary outcomes. Except for the number of falls which remained relatively stable, there was an improvement in all outcomes across both groups. However, these improvements were accompanied by moderate to high variability in measures such as dynamic balance, walking speed, functional and dual-task mobility, and the ability to safely balance. Therefore, interpretations of the findings should be approached cautiously. Our results align with those of a quasi-RCT involving 19 patients with PSP, which explored the effects of balance training plus eye movement training as compared to balance exercises alone on gait parameters, functional mobility and walking speed [19]. Results revealed improvements in stance time and walking speed after the combined balance and eye training and improvements in step length after balance training, with no between-group differences observed [19]. The effect sizes reported by the authors of that study were generally larger, differing from our findings in this aspect. Differences between our study and theirs may stem from the settings: ours occurred in an inpatient rehabilitation setting with multidisciplinary rehabilitation, whereas theirs was in a motion analysis laboratory without additional therapies. Additionally, it needs to be considered that despite both PD and PSP are neurodegenerative diseases affecting the basal ganglia and related structures within the extrapyramidal motor system, there are significant differences between the two, with PSP typically presenting with early and prominent problems related to balance and eye movement, particularly vertical gaze [51]. In addition, PSP often leads to more significant cognitive and behavioural changes [51]. Furthermore, there are differences between our intervention and the approach detailed in the eye movement training study conducted on patients with PSP. While their intervention encompassed conventional balance exercises, scanning exercises to enhance awareness, and computer-assisted saccade exercises that involved responding to randomly located visual stimuli on a computer screen, our intervention encompassed both eye movement training using custom-designed posters and wall markings, as well as a fusion of balance, coordination, strength, and walking exercises alongside eye movement training facilitated by wall markings. Drawing from insights in motor learning literature [52], we adopted this integrated and activity- and task-oriented methodology, aiming to present participants with tailored, pertinent, personalised, and modifiable motor tasks.

Eye movement analysis during FGA item No. 5 "pivot turn during walking" showed that participants were not able to fully direct their gaze towards the turning direction as their visual

focus did not extend to the farthest right visual region. We did not find relevant differences between time-points or groups in terms of the anticipated gaze direction during the turning manoeuvre, however. Our findings are consistent with previous research that has indicated similar patterns in steering control during turns among individuals with PD, showing an en bloc movement pattern, which contrasts with the body segment coordination patterns typically observed in healthy adults [53–55]. With respect to directing gaze towards the ground, after the intervention, participants in the AOPT-E group exhibited a decrease in the maximum fixation duration, whereas those in the AOPT group displayed an augmentation in the maximum fixation duration. Our findings contribute to these earlier discoveries by suggesting that shifting gaze towards the ground could potentially serve as a compensatory strategy in patients with PD [56] who cannot execute turns utilising anticipatory eye movements and coordinated body motions [53, 55]. This was confirmed by FGA item No. 6 results involving "stepping over an obstacle", indicating that the AOPT group maintained a relatively consistent level of fixations post-intervention. In contrast, the AOPT-E group exhibited a reduction in both the duration and number of fixations on the ground prior to the stepping-over action, while displaying an increased number of fixations on the object and the ground behind it before the action. Consistent with findings that individuals with PD, especially those experiencing FOG, tend to focus more on nearby areas of the ground and less on the target destination [56], our participants in the AOPT-E group demonstrated a reduced frequency of gazing towards the ground while performing the obstacle-crossing task. This suggests an expanded visual field, improvement in voluntary saccadic hypometria and improved dynamic balance during walking.

Analysing posters among all participants in the AOPT-E group revealed a reduction in the number of fixations from baseline to post-intervention, accompanied by extended fixation durations along the poster borders (edges, maximum right and left, top and bottom). This corresponds with alterations observed in eye movements during FGA tasks and indicates an expanded peripheral vision and overall visual field, improvement in voluntary saccadic hypometria, and enhanced control over eye movements. This effect was even more pronounced when examining the mixed poster, where the sequence of eye movements was randomised. Given the absence of comparable studies involving patients with PD, we draw insights from other clinical populations. The development of the eye movement training was based on previous studies, among one used a home-based balance and eye movement training intervention in patients with multiple sclerosis (MS) [31]. Comparable to our study, their results demonstrated improvements in balance, impact of MS on physical and psychological functions, and HRQoL. While their study did not evaluate effect sizes, it revealed low variability in these outcomes; combined with a larger sample size of 88 participants, this indicates that results can be generalised to ambulatory people with MS. Interventions in both our and their study involved static and dynamic balance and walking exercises, hand-eye coordination activities including ball tossing and catching, visual tracking and saccade training targeting stationary objects/dots in horizontal, vertical, and diagonal directions, visually fixating on immovable object/dots while moving the head/body, and individualised progression of difficulty.

A small study in patients post stroke investigated the effects of a 6-week eye movement training on walking speed, cadence, and step length, as compared to general gait training. Results showed improvements in all walking parameters specifically after eye movement training [57]. The eye movement intervention in this study encompassed tasks like visual search exercises using cards, reading upside down from turned cards, and maintaining focus on a moving baton's tip. Another study noted significant reductions in the fall risk among patients post stroke who participated in a 3-week home-based oculomotor and gaze stability exercise programme in conjunction with conventional rehabilitation, in contrast to those undergoing conventional rehabilitation alone [58]. Compared to our study, there was a large variability in

their dynamic balance and functional mobility findings. The eye movement training here entailed horizontal or vertical eye movements between fixed targets while keeping the head still, tracking a moving target using only the eyes while the head remains stationary, moving the head while maintaining gaze on a stationary target, and increasing the distance between the moving head and target (either horizontally or vertically) while tracking the target with the eyes. Interventions in these studies share some similarities with our standardised approach, as they encompassed non-technological, specific, and task-oriented methods, but these studies did not integrate components such as balance, coordination, strength, and walking training.

Our eye tracking results indicate that eye movement training shows promise in addressing FOG in people with PD. For instance, in FGA item 6, the AOPT-E group exhibited a remarkable shift in focus, reducing fixations on the ground before stepping over an obstacle and instead directing attention towards the obstacle itself and the surrounding terrain. This shift indicates improved balance and expanded visual awareness during walking, echoing recent findings that people with PD and FOG tend to concentrate more on immediate steps [59]. Furthermore, our study surpasses prior research revealing that people with PD exhibit fewer relevant fixations during regular walking but increase their focus on task-relevant elements during obstacle negotiation [60]. Cueing, known to aid walking in PD, especially for those with FOG [61] is supported by our results suggesting that eye movement training improves walking by increasing task-relevant fixation. Recent research underscores the profound impact of FOG on visual attention, with PD patients experiencing FOG showing prolonged fixation durations and a tendency to focus on nearby ground areas rather than their intended path, contrasting with the fixation patterns of those without FOG and healthy controls [56]. Our findings provide evidence for the potential benefits of AOPT-E for people with PD, particularly those with FOG. In FGA item No. 5, our analysis revealed a significant reduction in maximum fixation duration towards the ground post-intervention for the AOPT-E group, while the AOPT group exhibited the opposite trend. This suggests the potential efficacy of AOPT-E in modifying visual attention patterns, offering a promising avenue for enhancing mobility and potentially reducing FOG in people with PD.

In terms of feasibility, our analyses based on predefined criteria indicated that both the methods employed and a larger RCT are indeed feasible, notwithstanding recruitment delays stemming from COVID-19. Furthermore, regarding the acceptability of the AOPT-E intervention, our thematic analysis of interview data confirmed a high level of acceptability. Participants expressed satisfaction with the study's well-structured and diversified approach, offering an appropriate level of challenge. Participants perceived the training as intense, suggest that future sessions could benefit from a more balanced approach, potentially incorporating rest days.

Participants prioritised targeted patient education, a home exercise programme, and feedback from therapists. Goals centred around enhancing walking safety, developing coping strategies, offering active training options, scheduling preferred training times, and fostering a positive patient-therapist relationship. They reported finding the intervention enjoyable and meaningful, noting enhancements in activities of daily living. Overall, the feedback suggests that eye movement training holds promise as a feasible and potentially beneficial intervention for people with PD, warranting further exploration. To our knowledge, this study is the first to assess the preliminary effects of AOPT-E on HRQoL, FOG, and falls-related self-efficacy in patients with PD, while the results align with those from previous studies in patients with PSP, MS, and stroke.

## Study limitations

This study is not without limitations. First, our assessment focused on quantifying the number and duration of fixations within predetermined visual field zones, rather than measuring

saccades directly. We recognise this as a constraint. Our decision was guided by the assumption that an increase in fixations at the periphery of the visual field (i.e., the poster edge) would correspond to larger voluntary saccades.

Second, in this small pilot study, we were not able to consider subgroups of patients with and without FOG. Given the existing evidence demonstrating variations in saccade latency and variability between patients with PD with and without FOG [14], it would be crucial to assess disparities in saccades and fixations between these subgroups in a larger RCT with pre-specified subgroup analysis. It would be imperative to gather pilot data on saccade latency and variability following AOPT-E in patients with PD experiencing FOG. This would allow to estimate effect sizes and variability, thus aiding in the sample size calculation for a subsequent RCT. Subgroups would include patients with PD with FOG and those without FOG. Conducting subgroup analyses inherently divides the sample size, reducing the statistical power for detecting effects within each subgroup [62]. Therefore, to ensure sufficient power for planned subgroup analyses, it becomes necessary to increase the total sample size. The extent of this increase depends on various factors unique to each subgroup, including effect sizes and variability.

Third, this study took place within an inpatient rehabilitation facility, where patients were supplemented with AOPT-E and AOPT alongside multidisciplinary rehabilitation. Although we recognise that the multidisciplinary approach likely impacted the study outcomes, we presume that this influence was consistent across both groups under investigation.

Fourth, our interviews exclusively focused on patients in the AOPT-E group. Although we agree to the potential benefits of comparing patients' statements across both groups, we hypothesise that patients with PD are already acquainted with AOPT as a conventional rehabilitation approach and are likely to readily embrace the treatment.

Finally, our study did not include measurement of three-dimensional kinematics of the lower extremity, trunk, and head using a motion capture system. While we acknowledge this as a constraint, we contend that previous studies have employed this technology [55], yielding accessible outcomes.

## Conclusions

Results from our pilot study showed preliminary efficacy of AOPT-E and AOPT in improving dynamic balance, walking speed, the ability to safely balance, and functional and dual task mobility, HRQoL, FOG, falls-related self-efficacy, and depression in patients with PD. Based on the primary outcome results, the sample size for a full-scale RCT was established at 332 participants. According to predetermined criteria, feasibility of a larger RCT was confirmed. Improvements in patients' eye movements were observed after AOPT-E, suggesting an expanded peripheral vision and overall visual field, and enhanced control over eye movements.

## Supporting information

**S1 File. CONSORT checklist.**
(DOCX)

**S2 File. Qualitative strand: Interview guide, methods, and results.**
(DOCX)

**S3 File. Eye movement training: Poster description and further results.**
(DOCX)

**S1 Table. Intervention chart.**
(DOCX)

**S2 Table. Changes in dynamic balance during walking compared between the two groups.**
(DOCX)

**S3 Table. Changes in walking speed, mobility, balance, and fall risk in the two groups.**
(DOCX)

**S4 Table. Changes in health-related quality of life, falls-related self-efficacy, freezing of gait, and depression in the two groups.**
(DOCX)

**S1 Data.**
(PDF)

**S2 Data.**
(PDF)

## Acknowledgments

We would like to warmly thank all the participants in this study, as well as the physiotherapists who provided the study administration and intervention: Markus Rendl, Mattias Trummer, Sandra Berchtold, Miriam Hausberger, Andreas Mühlbacher, MSc, Barbara Linert, Ursula Miller, Rutger Lange, Hilde Weisenhorn, and Bianca Slamik, MSc. Many thanks to Professor Markus Reindl for his valuable assistance with the graphics.

## Author Contributions

**Conceptualization:** Michaela Stampfer-Kountchev, Christian Brenneis, Barbara Seebacher.

**Data curation:** Linda Rausch, Barbara Seebacher.

**Formal analysis:** Sarah Mildner, Franziska Kübler, Linda Rausch, Barbara Seebacher.

**Investigation:** Isabella Hotz, Franziska Kübler, Michaela Stampfer-Kountchev, Johanna Panzl, Christian Brenneis.

**Methodology:** Christian Brenneis, Barbara Seebacher.

**Project administration:** Isabella Hotz, Barbara Seebacher.

**Supervision:** Michaela Stampfer-Kountchev, Christian Brenneis, Barbara Seebacher.

**Writing – original draft:** Sarah Mildner, Barbara Seebacher.

**Writing – review & editing:** Isabella Hotz, Franziska Kübler, Linda Rausch, Michaela Stampfer-Kountchev, Johanna Panzl, Christian Brenneis, Barbara Seebacher.

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
