## [Decision Letter · Decision Letter 0]

11 Mar 2024

PONE-D-23-31439Effects of activity-oriented physiotherapy with and without eye movement training on dynamic balance, functional mobility, and eye movements in patients with Parkinson’s disease: an assessor-blinded randomised controlled pilot trialPLOS ONE

Dear Dr. Seebacher,

Thank you for submitting your manuscript to PLOS ONE. After careful consideration, we feel that it has merit but does not fully meet PLOS ONE’s publication criteria as it currently stands. Therefore, we invite you to submit a revised version of the manuscript that addresses the points raised during the review process.

**As you can see, the reviewers were overall positive about this manuscript.  Each reviewer raised several comments/questions that will improve the overall clarity of the manuscript.  We look forward to receiving your revised manuscript.**==============================

We look forward to receiving your revised manuscript.

Kind regards,

Eric R. Anson

Academic Editor

PLOS ONE

Journal Requirements:

2. In the online submission form, you indicated that [Insert text from online submission form here]. 

Reviewers' comments:

Reviewer's Responses to Questions

**Comments to the Author**

1. Is the manuscript technically sound, and do the data support the conclusions?

Reviewer #1: Yes

Reviewer #2: Partly

Reviewer #3: Yes

2. Has the statistical analysis been performed appropriately and rigorously? 

Reviewer #1: Yes

Reviewer #2: N/A

Reviewer #3: Yes

3. Have the authors made all data underlying the findings in their manuscript fully available?

Reviewer #1: Yes

Reviewer #2: No

Reviewer #3: Yes

4. Is the manuscript presented in an intelligible fashion and written in standard English?

Reviewer #1: Yes

Reviewer #2: Yes

Reviewer #3: Yes

5. Review Comments to the Author

Reviewer #1: This is a well-written report of a small pilot study of eye movement and Parkinsons. The pilot study was well-designed and the statistical analysis thorough. Randomization was well described and allocation concealment was accomplished, although single blind. The authors did not try to oversell the results or overanalyze the small amount of data. They list the study limitations honestly. However, I was disappointed in the discussion. After a pilot study that is designed to determine if a full-scale study is feasible, I want to know what was learned, statistically, that will inform the future design of a clinical trial. What underlying assumptions can now be made about the sample size? Were the underlying parameters (variability, effect size, etc.) consistent with the other study that was done on progressive supranuclear palsy or other studies mentioned (albeit not on PD). The authors opine that they could not do subgroup analyses. Can they do them in a full-scale study? Would the sample size requirements be feasible? What subgroups? In other words, you did a feasibility study to inform you how to design a full-scale trial. Now I want to know how you would design a full-scale trial!

Reviewer #2: In this study, Mildner and colleagues perform a pilot single-blind randomized controlled trial to investigate the added effects of eye movement training to functional gait and balance training. The study is innovative as it embeds technology into clinical therapy for people with Parkinson’s, and this generates feedback from participants that will be very useful for researchers and therapists. Further, training studies over longer periods like this one take considerable effort, so I commend the authors for this! On the other hand there are some inconsistencies with the (pre) registration, and some design issues that may impact the interpretation of the results. Additional clarification and discussion of these issues will be helpful for the subsequent reader. Kindly find detailed comments below:

1. Line 47: The specific aspects of feasibility are not mentioned in the results, while they are included in the objectives.

2. Line 105: There are some inconsistencies with the study registration such as whether it is a single center or multicenter study. Further in the registration the target sample size is 46 and the total sample size is 37. Here the target sample size is said to be 34 and only 25 participants are reported. Kindly clarify.

3. Line 147: If the participants were receiving many hours of rehabilitation each day besides the half hour for this study, how can one infer that the improvements at the end of the study are attributable to AOPT?

4. Line 174: Was AOPT delivered in the same room or under the same light conditions as the AOPT-E group?

5. Line 188: The AOPT-E group received a training that required dual-tasking – attention to the exercise as well as to the eye movements. Was there a control in terms of dual-task training for the AOPT group as well? Could this impact the results?

6. Line 200: In the trial registration FGA item 10 was to be assessed with the eye tracker, however there is no mention of this.

7. Line 262: The way the results are written up, it sure sounds like the groups have been statistically compared. I was confused by phrases like the “superiority of the group”, and “between-group differences showed”, “improvements for AOPT-E were greater than AOPT”, and “statistical analyses showed”. If the groups were not statistically tested, I would temper these statements.

8. Line 262: I also find it strange that no effect sizes were collected – for the purpose of running a subsequent RCT, an estimate of the effect size between interventions would be helpful.

Reviewer #3: This small pilot study (n=25) investigates an extremely under-researched topic area in PD rehab and I commend the authors for exploring the concept of visual training in their study. The manuscript is well written and clearly describes the results of a study comparing activity orientated physiotherapy with and without eye movement training.

My comments and suggestions are listed below:

1. Line 31 – “Outcomes were assessed at baseline, post-intervention, and

follow-up…” – the methods paragraph of the abstract would benefit from some further punctuation and clarification of the follow-up timescale, including which outcomes were completed at follow-up and when the interviews were conducted.

2. I appreciate that there is often overlap between pilot and feasibility studies, therefore, it would be useful to maintain some consistency to the order and primary focus of your aims/objectives. For example, in your abstract you talk about describing changes in balance/gait etc and then refer to feasibility outcomes. In your background section, you refer to feasibility outcomes first and then outline hypotheses which you later state are not tested due to the pilot nature of the study. In your results section, you first address co-primary and secondary outcomes and then feasibility outcomes. A separate heading to highlight your feasibility analysis would also help direct the reader.

3. In addition to the previous point, I would question the inclusion of specific hypotheses in the background/introduction (lines 89-96) as you later acknowledge that hypothesis testing was not conducted due to the small scale/pilot nature of the study. You also state that you did not conduct hypothesis testing to assess treatment effects within or between groups (line 261), but in your results you do make reference to within group and between-group analyses (e.g. line 374 - ). The numbers in this study are low and therefore these results should be interpreted with caution.

4. An additional paragraph in the background might be useful to highlight the links between attention and saccade frequency/visual function/gait impairment in PD as this is emerging as a significant contributer to falls and falls risk.

5. Line 105 – “We intended to recruit 17 patients per group and due to the pilot

character of the study, no sample size calculation was performed” – acknowledging that no sample size calculation was performed, what was your initial sample size of 17 per group based on? ie. patient flow/previous studies etc

6. Line 147 – It would be useful here to clarify for the reader why your patients were in hospital receiving inpatient rehabilitation in the first place. Was this a specialist PD rehab ward? What were the indications for their admission?

7. Line 150 – “… the inpatient rehabilitation program ranged from 2,240 to 2,600 minutes” – unclear what these numbers refer to. Your intervention groups received 16 x 30 minutes = 8 hours of intervention. This is probably sufficient information for the reader.

8. Line 221 – A reference to justify your list of feasibility criteria would be useful here, particularly as you state that your recruitment rate did not meet the target (line 410). How did you arrive at your target rate?

9. Table 1 – please check your female:male ratio in column two (AOPT group) – should this be a ratio of 3:9?

10. You state that both groups received identical physiotherapy interventions (line 162), and refer to eye movement training being conducted in a darkened room from a static seated position (line 174). Did both the groups undertake the “static and dynamic balance and walking exercises, hand-eye coordination activities including ball tossing and catching” or was this only part of the eye movement training? Although you have included the TIDER summary in supplementary material, it might be useful to clarify exactly what the eye movement training consisted of in the main body of the manuscript.

11. The final paragraph of the discussion section (line 551 onwards) is more descriptive than discursive. Given your findings, this section could be strengthened, for example, with reference to the potential impact of eye movement training on FOG , acknowledging existing literature in this field.

12. There are a lot of qualitative data relating to the interview responses in the Supplementary files – while I acknowledge these data are not the focal point of this study, the “five themes” (line 540) could be more clearly presented (i.e. as it reads presently, it is hard to determine what the five themes actually are ? training organisation; variety and challenge; training intensity; rest; targeted patient education; performance feedback; walking safety; strategies for optimal training; therapeutic relationship; pleasure and meaning). (While the S1 file is comprehensive, the data are also a little confusing in that the themes read more like statements or key findings).

6. PLOS authors have the option to publish the peer review history of their article (what does this mean?). If published, this will include your full peer review and any attached files.

Reviewer #1: No

Reviewer #2: **Yes: **Nicholas D'Cruz

Reviewer #3: No

---

## [Author Response · Author response to Decision Letter 0]

30 Mar 2024

PONE-D-23-31439

Effects of activity-oriented physiotherapy with and without eye movement training on dynamic balance, functional mobility, and eye movements in patients with Parkinson’s disease: an assessor-blinded randomised controlled pilot trial

Dear Dr. Seebacher,

Thank you for submitting your manuscript to PLOS ONE. After careful consideration, we feel that it has merit but does not fully meet PLOS ONE’s publication criteria as it currently stands. Therefore, we invite you to submit a revised version of the manuscript that addresses the points raised during the review process.

As you can see, the reviewers were overall positive about this manuscript. Each reviewer raised several comments/questions that will improve the overall clarity of the manuscript. We look forward to receiving your revised manuscript.

Dear Editor,

Dear Reviewers,

We thank the Editor and Reviewers for their valuable comments and feedback and for giving us a chance to submit a revised manuscript. Please find below our point-by-point responses. The main manuscript as well as S1 to S4 tables have been amended accordingly, with any changes highlighted to allow the Editor and Reviewers to check the adaptations made. 

We thank the Editor and Reviewers for their time and effort and hope we have sufficiently responded to the Editor’s and Reviewers’ requests. 

With kind regards,

Barbara Seebacher, on behalf of all the authors

We look forward to receiving your revised manuscript.

Kind regards,

Eric R. Anson

Academic Editor

PLOS ONE

Reviewers' comments:

Reviewer #1: 

Dear Reviewer #1, 

Thank you for reviewing our manuscript, acknowledging its value, and for offering your suggestions to enhance the quality of our manuscript. Please find our point-by-point response as follows.

This is a well-written report of a small pilot study of eye movement and Parkinsons. The pilot study was well-designed and the statistical analysis thorough. Randomization was well described and allocation concealment was accomplished, although single blind. The authors did not try to oversell the results or overanalyze the small amount of data. They list the study limitations honestly. However, I was disappointed in the discussion. After a pilot study that is designed to determine if a full-scale study is feasible, I want to know what was learned, statistically, that will inform the future design of a clinical trial. 

Point 1: What underlying assumptions can now be made about the sample size? Were the underlying parameters (variability, effect size, etc.) consistent with the other study that was done on progressive supranuclear palsy or other studies mentioned (albeit not on PD).

Response:

Thank you for bringing up this relevant point. In response to your suggestions, we meticulously examined the data and incorporated a discussion on variability, along with a comparison to the comparable studies in patients with progressive supranuclear palsy, multiple sclerosis, and stroke within the discussion section. Additionally, we provided a sample size calculation for a full-scale RCT in the discussion.

Point 2: The authors opine that they could not do subgroup analyses. Can they do them in a full-scale study? Would the sample size requirements be feasible? What subgroups? In other words, you did a feasibility study to inform you how to design a full-scale trial. Now I want to know how you would design a full-scale trial!

Response:

Thank you for bringing up this important question. In response to your question and to provide clarity for the reader, we have incorporated the following information into the paragraph:

[…] it would be crucial to assess disparities in saccades and fixations between these subgroups in a larger RCT with pre-specified subgroup analysis. It would be imperative to gather pilot data on saccade latency and variability following AOPT-E in patients with PD experiencing FOG. This would allow to estimate effect sizes and variability, thus aiding in the sample size calculation for a subsequent RCT. Subgroups would include patients with PD with FOG and those without FOG. Conducting subgroup analyses inherently divides the sample size, reducing the statistical power for detecting effects within each subgroup [1]. Therefore, to ensure sufficient power for planned subgroup analyses, it becomes necessary to increase the total sample size. The extent of this increase depends on various factors unique to each subgroup, including effect sizes and variability.

Reviewer #2: 

Dear Reviewer #2,

Dear Dr D'Cruz,

Thank you for reviewing our manuscript, recognising its relevance, and for providing valuable suggestions to improve its quality. Please find our point-by-point response as follows.

In this study, Mildner and colleagues perform a pilot single-blind randomized controlled trial to investigate the added effects of eye movement training to functional gait and balance training. The study is innovative as it embeds technology into clinical therapy for people with Parkinson’s, and this generates feedback from participants that will be very useful for researchers and therapists. Further, training studies over longer periods like this one take considerable effort, so I commend the authors for this! On the other hand there are some inconsistencies with the (pre) registration, and some design issues that may impact the interpretation of the results. Additional clarification and discussion of these issues will be helpful for the subsequent reader. Kindly find detailed comments below:

Point 1. Line 47: The specific aspects of feasibility are not mentioned in the results, while they are included in the objectives.

Response:

Thank you for providing this feedback. The sole reason for the omission of detailed feasibility results in the abstract was the constraint of the word count. Following your recommendations and those of Reviewer #3, we have moderated our statements regarding the results of primary and secondary outcomes. While we initially did not include effect size calculations based on a methods paper [2], we have now incorporated them. As a result, we have thoroughly revised our abstract, results and discussion to include specifics on the feasibility results.

2. Line 105: There are some inconsistencies with the study registration such as whether it is a single center or multicenter study. Further in the registration the target sample size is 46 and the total sample size is 37. Here the target sample size is said to be 34 and only 25 participants are reported. Kindly clarify.

Response:

Thank you for bringing up this point. We acknowledge that there appear to be inconsistencies with the study registration, particularly concerning the inclusion of a small study within this study, which will be published separately (https://drks.de/search/en/trial/DRKS00024982). In the section regarding the 'Health condition or problem studied,' we provided the information that the secondary study pertains to expert interviews involving professionals from various medical fields, with no specific health issue addressed. Additionally, under 'Interventions, Observational Groups,' we detailed the parameters of the secondary study, specifying a group of 12 experts to be interviewed for 45 minutes each.

This was also the reason for the discrepant target sample size. For the present study in people with Parkinson’s disease, the target sample size was 34. For the secondary study, the target sample size was 12. The decision to consolidate the registration for both studies was made because the studies were related insofar, as the secondary study involved qualitative interviews with experts on the barriers and facilitators of RCTs in rehabilitation centres. The secondary study's topic emerged due to the challenging circumstances brought about by the COVID-19 pandemic, particularly the strain it placed on health institutions, including our rehabilitation centre. Under such conditions, preparing for a study was arduous, let alone recruiting patients and involving already short-staffed personnel. Consequently, these two studies became somewhat intertwined, despite addressing different topics. Looking back, it would have been more appropriate to register them separately. We trust that this explanation addresses any lingering uncertainties.

Regarding reporting only 25 participants, our Principal Investigator made the decision to conclude the study due to the challenging circumstances at our rehabilitation centre, compounded by staff shortages. We have revised our statement at the beginning of the results section to provide clarity for the reader. The updated statement now reads as follows: ‘Due to COVID-19 restrictions, safety measures, and staffing limitations, recruiting the intended 34 patients with PD was not feasible. Out of the 25 randomised participants, 24 successfully completed the study and were included in the analysis […].’

3. Line 147: If the participants were receiving many hours of rehabilitation each day besides the half hour for this study, how can one infer that the improvements at the end of the study are attributable to AOPT?

Response:

Thank you for commenting on this There exists a substantial body of literature focusing on interventions implemented alongside either inpatient or outpatient multidisciplinary rehabilitation, such as robot-assisted gait training in neurorehabilitation centres [3]. Moreover, numerous studies have explored the supplementary effects of specific interventions when combined with other therapies, such as the additional benefits of kinesiotape alongside acupuncture [4]. While we acknowledge that inpatient rehabilitation likely exerted an effect, it needs to be considered that both study groups received equal amounts of inpatient rehabilitation. In response to your input and in consideration of comparing our findings with those from a study involving individuals with PSP, we have included the following sentence in the discussion section. ‘Differences between our study and theirs may stem from the settings: ours occurred in an inpatient rehabilitation setting with multidisciplinary rehabilitation, whereas theirs was in a motion analysis laboratory without additional therapies.’

In addition, we added the following information as a study limitation: ‘Third, this study took place within an inpatient rehabilitation facility, where patients were supplemented with AOPT-E and AOPT alongside multidisciplinary rehabilitation. Although we recognise that the multidisciplinary approach likely impacted the study outcomes, we presume that this influence was consistent across both groups under investigation.’

4. Line 174: Was AOPT delivered in the same room or under the same light conditions as the AOPT-E group?

Response:

Yes, AOPT and AOPT-E were delivered in the same room or under the same light conditions. We removed the specific information for the AOPT-E group and incorporated the following details into the intervention section: ‘AOPT-E and AOPT were delivered in a darkened room with white light at 300 lux.’ In addition, we clarified the information in the TIDieR checklist.

5. Line 188: The AOPT-E group received a training that required dual-tasking – attention to the exercise as well as to the eye movements. Was there a control in terms of dual-task training for the AOPT group as well? Could this impact the results?

Response:

Thank you for pointing out this discrepancy. Reviewer #3 also raised a similar concern. We intended to clarify this in the TIDieR 'Materials' section, where we specified the use of various physiotherapy equipment such as chairs, stools, gym mats, gym balls, tennis balls, cones, ropes, step stools, dumbbells, balance boards, coordination hoops, rice, and bean bags.’

Both groups of participants underwent training involving either single-task or dual-task exercises. In fact, both groups engaged in static and dynamic balance and walking exercises, as well as hand-eye coordination activities such as ball tossing and catching. We have now added the information to the manuscript and updated the TIDieR checklist to reflect these corrections.

6. Line 200: In the trial registration FGA item 10 was to be assessed with the eye tracker, however there is no mention of this.

Response:

Thank you for bringing attention to this oversight. Regrettably, we omitted reporting the results concerning FGA item 10 due to technical complications that impeded our ability to obtain precise data. The suboptimal lighting conditions on the stairway and persistent challenges with calibration during participants' stair climbing contributed to inaccuracies in the recorded gaze data. Consequently, conducting meaningful data analysis was not feasible. To improve transparency in our reporting, we have now incorporated FGA item 10 into the primary outcomes section. Furthermore, we have detailed the encountered problems in the results section for clarity: ‘Regarding FGA item 10, we faced technical challenges that hindered our ability to gather accurate data. Inadequate lighting conditions along the stairway and ongoing difficulties with calibration during participants' stair climbing resulted in inaccuracies in the recorded gaze data. As a result, conducting a meaningful data analysis was not possible.’

7. Line 262: The way the results are written up, it sure sounds like the groups have been statistically compared. I was confused by phrases like the “superiority of the group”, and “between-group differences showed”, “improvements for AOPT-E were greater than AOPT”, and “statistical analyses showed”. If the groups were not statistically tested, I would temper these statements.

Response:

We greatly appreciate this feedback. Reviewer #3 also expressed similar concerns. In response, we have revised our description throughout the manuscript to reflect the fact that the groups were not statistically compared.

8. Line 262: I also find it strange that no effect sizes were collected – for the purpose of running a subsequent RCT, an estimate of the effect size between interventions would be helpful.

Response:

Once more, we sincerely value this feedback. Reviewer #1 shared similar suggestions. Consequently, we have incorporated details on the estimation of effect sizes in the data analysis section. Furthermore, we have integrated effect sizes into Tables 2-4. Utilising the effect sizes derived from the primary outcome results, we determined the sample size for a full-scale RCT, with this information now included in the discussion.

Reviewer #3: 

Dear Reviewer #3,

Thank you for reviewing our manuscript, recognising its relevance, and for providing valuable suggestions to improve its quality. Please find our point-by-point response as follows.

This small pilot study (n=25) investigates an extremely under-researched topic area in PD rehab and I commend the authors for exploring the concept of visual training in their study. The manuscript is well written and clearly describes the results of a study comparing activity orientated physiotherapy with and without eye movement training.

My comments and suggestions are listed below:

1. Line 31 – “Outcomes were assessed at baseline, post-intervention, and follow-up…” – the methods paragraph of the abstract would benefit from some further punctuation and clarification of the follow-up timescale, including which outcomes were completed at follow-up and when the interviews were conducted.

Response:

We appreciate your comment. Based on it, we have adjusted the methods section of the abstract, now stating: ‘Outcomes were assessed at baseline and post-intervention, including dynamic balance, walking speed, functional and dual-task mobility, ability to safely balance, health-related quality of life (HRQoL), depression, and eye movements (number and duration of fixations) using a mobile eye tracker. Freezing of gait (FOG), and falls-related self-efficacy were assessed at baseline, post-intervention, and 4-week follow-up. Feasibility was assessed using predefined criteria including recruitment, retention and adherence rates, adverse events, falls, and acceptability of the intervention, the latter evaluated at post-intervention utilising qualitative interviews.’

2. I appreciate that there is often overlap between pilot and feasibility studies, therefore, it 

---

## [Decision Letter · Decision Letter 1]

14 May 2024

PONE-D-23-31439R1Effects of activity-oriented physiotherapy with and without eye movement training on dynamic balance, functional mobility, and eye movements in patients with Parkinson’s disease: an assessor-blinded randomised controlled pilot trialPLOS ONE

Dear Dr. Seebacher,

Thank you for submitting your manuscript to PLOS ONE. After careful consideration, we feel that it has merit but does not fully meet PLOS ONE’s publication criteria as it currently stands. Therefore, we invite you to submit a revised version of the manuscript that addresses the points raised during the review process.

Overall the reviewers were very positive concerning this revision.  There is only 1 minor revision requested at this time and I believe that if this is addressed within the methods paragraph on statistics that will be sufficient.  Please also provide a citation justifying the effect size breakdown. "The only minor comment I have is in relation to your use of effect sizes in the revised manuscript- I wonder if it might be useful to add a little clarity in your abstract/statistics paragraph to help the reader interpret what your r values mean e.g. small/medium/large effect sizes."On receipt of this minor revision, your manuscript will be acceptable for publication at PLOS. Please submit your revised manuscript by Jun 28 2024 11:59PM. If you will need more time than this to complete your revisions, please reply to this message or contact the journal office at plosone@plos.org. Please include the following items when submitting your revised manuscript:A rebuttal letter that responds to each point raised by the academic editor and reviewer(s). You should upload this letter as a separate file labeled 'Response to Reviewers'.A marked-up copy of your manuscript that highlights changes made to the original version. You should upload this as a separate file labeled 'Revised Manuscript with Track Changes'.An unmarked version of your revised paper without tracked changes. You should upload this as a separate file labeled 'Manuscript'.If applicable, we recommend that you deposit your laboratory protocols in protocols.io to enhance the reproducibility of your results. Protocols.io assigns your protocol its own identifier (DOI) so that it can be cited independently in the future. For instructions see: https://journals.plos.org/plosone/s/submission-guidelines#loc-laboratory-protocols. Additionally, PLOS ONE offers an option for publishing peer-reviewed Lab Protocol articles, which describe protocols hosted on protocols.io. Read more information on sharing protocols at https://plos.org/protocols?utm_medium=editorial-email&utm_source=authorletters&utm_campaign=protocols.

We look forward to receiving your revised manuscript.

Kind regards,

Eric R. Anson

Academic Editor

PLOS ONE

Journal Requirements:

Reviewers' comments:

Reviewer's Responses to Questions

**Comments to the Author**

1. If the authors have adequately addressed your comments raised in a previous round of review and you feel that this manuscript is now acceptable for publication, you may indicate that here to bypass the “Comments to the Author” section, enter your conflict of interest statement in the “Confidential to Editor” section, and submit your "Accept" recommendation.

Reviewer #1: All comments have been addressed

Reviewer #2: All comments have been addressed

Reviewer #3: (No Response)

2. Is the manuscript technically sound, and do the data support the conclusions?

Reviewer #1: (No Response)

Reviewer #2: Yes

Reviewer #3: Yes

3. Has the statistical analysis been performed appropriately and rigorously? 

Reviewer #1: (No Response)

Reviewer #2: Yes

Reviewer #3: I Don't Know

4. Have the authors made all data underlying the findings in their manuscript fully available?

Reviewer #1: (No Response)

Reviewer #2: No

Reviewer #3: Yes

5. Is the manuscript presented in an intelligible fashion and written in standard English?

Reviewer #1: (No Response)

Reviewer #2: Yes

Reviewer #3: Yes

6. Review Comments to the Author

Reviewer #1: (No Response)

Reviewer #2: Thank you for the clear and transparent responses to the comments and for the adaptations to the manuscript. This work now seems fit for publication. Congratulations, and all the best!

Reviewer #3: Thank you for your detailed response to the feedback from reviewers and providing clarity on the comments I raised. I have enjoyed reading your paper in its revised format and I think the discussion is much stronger now. I wish you all the best for your future research in this exciting area.

The only minor comment I have is in relation to your use of effect sizes in the revised manuscript- I wonder if it might be useful to add a little clarity in your abstract/statistics paragraph to help the reader interpret what your r values mean e.g. small/medium/large effect sizes.

7. PLOS authors have the option to publish the peer review history of their article (what does this mean?). If published, this will include your full peer review and any attached files.

Reviewer #1: No

Reviewer #2: **Yes: **Nicholas D'Cruz

Reviewer #3: No

---

## [Author Response · Author response to Decision Letter 1]

16 May 2024

PONE-D-23-31439R1

Effects of activity-oriented physiotherapy with and without eye movement training on dynamic balance, functional mobility, and eye movements in patients with Parkinson’s disease: an assessor-blinded randomised controlled pilot trial

Dear Dr. Seebacher,

Thank you for submitting your manuscript to PLOS ONE. After careful consideration, we feel that it has merit but does not fully meet PLOS ONE’s publication criteria as it currently stands. Therefore, we invite you to submit a revised version of the manuscript that addresses the points raised during the review process.

Overall the reviewers were very positive concerning this revision. There is only 1 minor revision requested at this time and I believe that if this is addressed within the methods paragraph on statistics that will be sufficient. Please also provide a citation justifying the effect size breakdown. "The only minor comment I have is in relation to your use of effect sizes in the revised manuscript- I wonder if it might be useful to add a little clarity in your abstract/statistics paragraph to help the reader interpret what your r values mean e.g. small/medium/large effect sizes."

On receipt of this minor revision, your manuscript will be acceptable for publication at PLOS.

Dear Editor,

Dear Reviewers,

We thank the Editor and Reviewer #3 for their valuable minor comment and feedback and for giving us a chance to submit a revised manuscript. Please find below our point-by-point responses. The main manuscript has been amended accordingly, with any changes highlighted to allow the Editor and Reviewers to check the adaptations made. 

We thank the Editor and Reviewers for their time and effort and hope we have sufficiently responded to the Editor’s and Reviewers’ requests. 

With kind regards,

Barbara Seebacher, on behalf of all the authors

We look forward to receiving your revised manuscript.

Kind regards,

Eric R. Anson

Academic Editor

PLOS ONE

Review Comments to the Author

Reviewer #1: (No Response)

Dear Reviewer #1, Thank you for your previous valuable suggestions and for acknowledging the changes we made.

Reviewer #2: Thank you for the clear and transparent responses to the comments and for the adaptations to the manuscript. This work now seems fit for publication. Congratulations, and all the best!

Dear Reviewer #2, Thank you for your previous valuable suggestions, for acknowledging the changes we made, and for your good wishes.

Reviewer #3: Thank you for your detailed response to the feedback from reviewers and providing clarity on the comments I raised. I have enjoyed reading your paper in its revised format and I think the discussion is much stronger now. I wish you all the best for your future research in this exciting area.

Dear Reviewer #3, Thank you for your previous valuable suggestions, for acknowledging the changes we made, and best wishes.

Point 1: The only minor comment I have is in relation to your use of effect sizes in the revised manuscript- I wonder if it might be useful to add a little clarity in your abstract/statistics paragraph to help the reader interpret what your r values mean e.g. small/medium/large effect sizes.

Response 1: Thank you for your advice. We have now included the following sentence in the abstract: ‘Effect sizes of 0.10 were considered weak, 0.30 moderate, and ≥0.50 strong.’ In the statistical analysis section, we have included the following statement: ‘According to Cohen (1988), a correlation coefficient of 0.10 is considered weak, 0.30 is considered moderate, and 0.50 or higher is considered strong. [1].’

Reference

1. Cohen J. Statistical power analysis for the behavioral sciences. 2nd ed ed. Hillsdale: Lawrence Erlbaum Associates; 1988.

---

## [Editor Report · Decision Letter 2]

20 May 2024

Effects of activity-oriented physiotherapy with and without eye movement training on dynamic balance, functional mobility, and eye movements in patients with Parkinson’s disease: an assessor-blinded randomised controlled pilot trial

PONE-D-23-31439R2

Dear Dr. Seebacher,

We’re pleased to inform you that your manuscript has been judged scientifically suitable for publication and will be formally accepted for publication once it meets all outstanding technical requirements.

Kind regards,

Eric R. Anson

Academic Editor

PLOS ONE
---

## [Editor Report · Acceptance letter]

24 May 2024

PONE-D-23-31439R2 

PLOS ONE

Dear Dr. Seebacher, 

I'm pleased to inform you that your manuscript has been deemed suitable for publication in PLOS ONE. Congratulations! Your manuscript is now being handed over to our production team.

Kind regards, 

on behalf of

Dr. Eric R. Anson 

Academic Editor

PLOS ONE